# Family cohesion predicts long-term health and well-being after losing a parent to cancer as a teenager: A nationwide population-based study

**Dröfn Birgisdóttir**[1]*, **Tove Bylund Grenklo**[2], **Ulrika Kreicbergs**[3,4], **Gunnar Steineck**[5,6], **Carl Johan Fürst**[1], **Jimmie Kristensson**[7]

1 Faculty of Medicine, Department of Clinical Sciences Lund, The Institute for Palliative Care, Respiratory Medicine, Allergology, and Palliative Medicine, Lund University, Lund, Sweden, 2 Faculty of Health and Occupational Studies, Department of Caring Science, University of Gävle, Gävle, Sweden, 3 Department of Caring Sciences, Palliative Research Centre, Ersta Sköndal Bräcke University College, Stockholm, Sweden, 4 Department of Women's and Children's Health, Karolinska Institutet, Stockholm, Sweden, 5 Department of Oncology-Pathology, Division of Clinical Cancer Epidemiology, Karolinska Institute, Stockholm, Sweden, 6 Department of Oncology, Division of Clinical Cancer Epidemiology, Institute of Clinical Sciences, Sahlgrenska Academy at the University of Gothenburg, Gothenburg, Sweden, 7 Faculty of Medicine, Department of Health Sciences, Lund University, Lund, Sweden

* drofn.birgisdottir@med.lu.se

## Abstract

### Background

Parentally bereaved children are at increased risk of negative consequences, and the mediating factors most consistently identified are found to be related to family function after the loss, including cohesion. However, existing evidence is limited, especially with respect to children and youths' own perception of family cohesion and its long-term effects on health and well-being. Therefore, the aim of this study was to investigate self-reported family cohesion the first year after the loss of a parent to cancer and its association to long-term psychological health and well-being among young adults that were bereaved during their teenage years.

### Method and participants

In this nationwide population-based study, 622 of 851 (73%) young adults (aged 18–26) responded to a study-specific questionnaire six to nine years after losing a parent to cancer at the age of 13 to 16. Associations were assessed with modified Poisson regression.

### Results

Bereaved youth that reported poor family cohesion the first year after losing a parent to cancer had a higher risk of reporting symptoms of moderate to severe depression six to nine years after the loss compared to those reporting good family cohesion. They also had a higher risk of reporting low levels of well-being, symptoms of anxiety, problematic sleeping

**Data Availability Statement:** In order to assure data confidentiality and to protect the privacy of the research participants, the datasets generated and/

or analyzed during the current study are not publicly available due to legal and ethical restrictions as described by the Swedish law (2003:460) and the Swedish Ethical Review Authority (https://etikprovningsmyndigheten.se) regarding processing of sensitive data. Data can be made available from the Research Data Office, Data Access Unit, at the Karolinska Institute, Stockholm, Sweden (contact via rdo@ki.se), for researchers who meet the criteria for access to confidential data.

**Funding:** The Swedish Cancer Foundation (2008–758), https://www.cancerfonden.se (GS); The Kamprad Family Foundation for Entrepreneurship, https://familjenkampradsstiftelse.se/in-english/ (CJ); The Mats Paulsson Foundation, https://www.matspaulssonstiftelserna.com (CJ), and the Gålö Foundation, https://www.galostiftelsen.se (UK) supported the research project. The funders had no role in study design, data collection and analysis, decision to publish, or preparation of the manuscript.

**Competing interests:** The authors have declared that no competing interests exist.

**Abbreviations:** PHQ-9, The Patient Health Questionnaire-9 scale; RR, Risk ratio; CI, Confidence interval.

and emotional numbness once a week or more at the time of the survey. These results remained statistically significant after adjusting for a variety of possible confounding factors.

## Conclusion

Self-reported poor family cohesion the first year after the loss of a parent to cancer was strongly associated with long-term negative psychological health-related outcomes among bereaved youth. To pay attention to family cohesion and, if needed, to provide support to strengthen family cohesion in families facing bereavement might prevent long-term suffering for their teenage children.

## Introduction

The early loss of a parent to death during childhood places significant stress on children and adolescents [1–3] and has been shown to have various effects on the health and well-being of children and adolescents, including higher risk of depression [4–6], anxiety [5–10], suicide attempts [11,12] and self-injurious behaviors [13,14], compared to their non-bereaved peers. The factors most consistently found to affect the health and well-being of children and adolescents, in general as well as in bereavement, relate to family functioning [7,15,16].

The relationship between the surviving parent and the bereaved child seems to play a big role [17,18], including warmth, emotional connection and positive parenting [15,19–21], as does the mental health of the surviving parent [9,15,16,22,23]. The concept of family can be described as a complex social unit, that includes members who interact with each other and influence each other's behaviors [24], in a way that can be both supportive or disruptive to the wellbeing of the individuals in the family [25]. One of the central aspects of family function is family cohesion [26].

Family cohesion is a broad concept that captures the emotional bonds between family members, the feeling of togetherness, along with support, intimacy and time spent together. Family cohesion can vary over time and is highly influenced by external factors and strains, as well as from factors within the family [26]. It has been suggested that more studies are needed to identify subgroups of bereaved children and adolescents that are at risk of developing mental health problems [27] or are in need of bereavement support [28–30]. When it comes to teenagers and youths' own perception of family cohesion after the loss of a parent and its relation to long-term health and well-being, the literature is limited.

## Aim

The aim of this study was to investigate the level of self-reported family cohesion the first year after the death of a parent and its association with long-term: quality of life; well-being; depressive symptoms; symptoms of anxiety; emotional numbness; and problematic sleeping, in young adults (age 18–26) who lost their parent to cancer at the age of 13–16.

## Method

This study is a part of a Swedish nationwide population-based study on young adults who lost a parent to cancer as teenagers [13,31,32]. The Regional Ethical Review Board of Karolinska Institute, Stockholm, Sweden, approved the study (2007/836-31).

## Participants

Individuals who had died from cancer (based on International Classification of Diseases, 10th revision (ICD-10), codes C00–C96) before the age of 65, in the years 2000 to 2003, were identified in the Swedish National Causes of Death Register. The register holders at the Multi-generation register, at Statistics Sweden, then used this information to identify young adults who had lost a parent between the ages of 13 and 16, which is equivalent to the age of lower secondary school (or junior high school). The decision to narrow the focus of the study to this age span was made to account for the great variation in levels of maturity that occurs throughout the teenage years [33]. Furthermore, for inclusion in this study, the participants also needed to have been born in one of the Nordic countries, have the same registered address as both parents at the time of death, and have the surviving parent alive at the time of follow-up, which took place in 2009 to 2010. Furthermore, all participants needed to be living in Sweden at the time of the study, have identifiable telephone numbers, and be able to read, write and understand Swedish.

## Ethical approval and consent to participate

The research was performed in accordance with the Helsinki Declaration and was reviewed by the Regional Ethics Review Board at the Karolinska Institute (2007/836–31), which gave approval for the research to be carried out. Each participant was informed both orally and in writing about the aim of the study, as well as their right to withdraw from the study at any time. Each participant gave informed consent for their participation as required by Swedish Ethical Review Authority and in accordance with Swedish law.

## Questionnaire development

A study-specific questionnaire was used to collect the data. The development of the questionnaire followed well established routines described in previous articles [34,35]. The content and wording of the questions were based on topics identified in semi-structured interviews with 15 cancer-bereaved youth (aged 13–25) as well as previous questionnaires from the research group, the bereavement literature, and interviews with three experts specialized in palliative care and bereavement. Face validity was tested in think-aloud interviews with six of the previously interviewed individuals and nine new participants [13]. After final adjustments the questionnaire included 271 question items, asking about the participants' background, experiences before and after the death of their parent, as well as their current health and well-being. A total of 36 question items were considered relevant for this study, namely family cohesion variables (n = 2), health-related outcome variables (n = 18), potential confounding variables (n = 10) and other background variables (n = 6).

## Measurements

The perception of family cohesion was evaluated with the following question:

Did you as a family have good cohesion during:

   a) your childhood (until you were approximately 11–12 years old)?

   b) your teenage years (until the death of your parent)?

   c) 0–6 months after your loss?

   d) 7–12 months after your loss?

   e) today?

For each sub-question, the participants were able to choose from four response alternatives and family cohesion was labelled poor if the response was "No, not at all" or "Yes, a little", while "Yes, moderate" and "Yes, very good" were labelled as good family cohesion.

To use as an exposure variable in this study, sub-questions c) and d) were combined into one variable labelled "*Self-reported family cohesion the first year after the loss*" and those reporting poor family cohesion 0–6 and/or 7–12 months after the loss of a parent constituted the exposure group.

Depression was measured using the Patient Health Questionnaire-9 scale (PHQ-9), with the cut-off set to ≥10 points to indicate moderate to severe depression. As described in a systematic review of the Patient Health Questionnaire scales, the PHQ-9 is an efficient and valid scale to use for detection, differentiation and monitoring of depression [36].

Following the approach "one phenomenon–one direct question", in which the question is related to a real-life phenomenon [37], the outcomes of well-being, quality of life and emotional numbness were measured with the following single-item questions: "Have you experienced high levels of well-being in the last month?"; "Have you had a good quality of life the last month?"; and "Have you felt emotionally numb (cut off, like you were in a bubble or had a wall around you) in the last month?". Four single-item questions, including "Have you felt persistent worries (fear, anxiety) in the last month?" and "Have you had sudden attacks of anxiety (fear) in the last month?", were used to measure symptoms of anxiety. Problematic sleeping was measured with two questions, one of which was "Have you had trouble falling asleep at night in the last month?"

The questions used for the six outcome measurements can be seen together with the response alternatives and their categorization in S1 Table.

## Data collection

Each participant that met the inclusion criteria received an invitation letter with information about the project, followed by an informative telephone call from a research assistant. If oral consent to participate in the study was given, the questionnaire and an ethics information sheet along with a separate reply card was sent out. Information about their right to withdraw from the study at any time was given both orally and in writing. To ensure anonymity, the questionnaires were returned in pre-stamped envelopes, separately from the reply cards. To minimize the risk of causing distress to participants, questions were carefully phrased and data collection was avoided during holiday seasons and the anniversaries of parental loss.

## Statistical analyses

Associations between self-reported family cohesion the first year after the death of a parent and the six health-related outcome variables were assessed using modified Poisson regression (with robust variance) [38]. The results are presented as relative risk ratios (RR) with corresponding confidence intervals (CI) set at 95%. Individuals with missing values were excluded from the statistical analysis for each calculation.

For the purpose of minimizing the risk of systematic errors [37,39] the literature was searched for risk factors, related to family cohesion or to any of the outcome variables, that could be considered as possible confounding variables. After a discussion within the research group, based on the literature, previous research from the group, and with the help of directed acyclic graphs, 10 possible confounding variables were selected and used to build a multivariable Poisson regression model. Initially, the possible confounders were each classified as either "Background variables", "Family-related variables" or variables regarding "Adverse events

during childhood and the awareness time of parent's imminent death". The possible con-founding variables, their categories and prevalence can be seen in Table 1.

For each of the outcome variables, adjusted RRs were calculated with the model in three steps, subsequently adjusting for the three groups of possible confounders. Further analysis was also made, in which crude and adjusted RRs were re-calculated, as described above, but now with the data stratified by the gender of the deceased parent.

## Results

A total of 851 cancer-bereaved young adults, aged 18 to 26, were eligible for participation in the study, 6.5 percent were not reachable, 7.8 percent declined participation and 12.7 percent did not return the questionnaire. In the end, 622 individuals answered and returned the questionnaire (73% response rate), of which 49.8% were male and 50.2% female. Table 2 presents an overview of the study participant characteristics.

In total, 134 (21.7%) of the parentally bereaved participants experienced family cohesion as poor (no/little) at some point during the first year after the death of a parent while 483 (78.3%) reported good (moderate/very good) family cohesion during the same period (missing values: 5 (0.8%)).

Among the cancer-bereaved young adults reporting poor family cohesion the first year after the loss of a parent, 31.6% reported moderate to severe depression according to the PHQ-9 depression scale at the time of the survey (six to nine years after the loss of the parent) compared to 8.6% of those reporting good family cohesion in the first year post loss. Also, for those reporting poor family cohesion, 36.4% reported low levels of well-being compared to 15.9% of those reporting good family cohesion. For all the health-related indicators under investigation a higher prevalence was seen among those who reported poor family cohesion the first year after the loss, compared to those who reported good family cohesion during the same period (Fig 1).

Table 3 shows the crude and adjusted RRs, with corresponding 95% CI, for the long-term negative health-related outcomes in relation to the self-reported family cohesion the first year after the loss of a parent to cancer. The cancer-bereaved young adults that reported poor family cohesion the first year after the death of a parent had a higher risk of reporting symptoms of moderate to severe depression in the last two weeks, compared to those reporting good family cohesion (RR: 3.67, 95%CI: 2.50–5.40). They were also more likely to report low well-being (RR: 2.28, 95%CI: 1.68–3.10), symptoms of anxiety (RR: 2.32, 95%CI: 1.64–3.29), problematic sleeping (RR: 1.80 95%CI: 1.30–2.48) and emotional numbness (RR: 2.86, 95%CI: 1.91–4.30) once a week or more in the last month, compared to those reporting good family cohesion the first year after the loss. All these results stayed statistically significant after adjusting for possible confounders in the three-step multivariable modified Poisson regression model (Table 3). In addition, those reporting poor family cohesion the first year after the loss were also more likely to report low quality of life (RR: 1.50 95%CI: 1.02–2.22) at the time of the survey, but this difference did not remain statistically significant in the adjustments for possible confounding factors (Table 3).

Further analysis of the data was also performed with the data stratified by the gender of the diseased parent. Both crude and adjusted RRs were calculated again using the same three groups of possible confounders as before, with the results presented in Table 3. In both the paternally and maternally bereaved participants, a higher risk of long-term moderate to severe depression was seen in those that had experienced poor family cohesion the first year after loss compared to those that had reported good family cohesion. However, a higher risk of reporting symptoms of anxiety and emotional numbness was statistically significant only among the

**Table 1. Possible confounding variables.**

| Possible confounding variables used in the multivariable modified Poisson regression model | Poor[a] family cohesion the first year after the loss n (%) | Good[b] family cohesion the first year after the loss n (%) | Total n |
|---|---|---|---|
| **Step 1—Background variables** | | | |
| Gender of the participants | | | |
| Male | 39 (13) | 266 (87) | 305 |
| Female | 95 (30.5) | 216 (69.5) | 311 |
| Age at loss | | | |
| 13 | 23 (19) | 98 (81) | 121 |
| 14 | 45 (29) | 112 (71) | 157 |
| 15 | 31 (18.5) | 137 (81.5) | 168 |
| 16 | 33 (21) | 124 (79) | 157 |
| Year of birth | | | |
| 1988–1990 | 48 (23) | 161 (77) | 209 |
| 1986–1987 | 58 (20) | 226 (80) | 284 |
| 1984–1985 | 28 (23) | 93 (77) | 121 |
| **Step 2—Family-related variables** | | | |
| Depression in at least one parent | | | |
| Yes | 34 (34) | 66 (66) | 100 |
| No | 95 (19) | 414 (81) | 509 |
| Number of siblings | | | |
| 0 | 6 (17) | 30 (83.3) | 36 |
| 1 | 65 (26.5) | 180 (73.5) | 245 |
| 2 | 47 (24) | 148 (76) | 195 |
| 3 or more | 16 (12) | 121 (88) | 137 |
| Education level of surviving parent | | | |
| Middle school (≤ 9th grade) | 18 (16) | 94 (84) | 112 |
| High school (≥10th grade) | 54 (22) | 194 (78) | 248 |
| College/university | 56 (25) | 170 (75) | 226 |
| Alcohol and/or drug misuse in at least one parent | | | |
| Yes | 12 (29) | 29 (71) | 41 |
| No | 119 (21) | 452 (79) | 571 |
| **Step 3—Adverse events during childhood and awareness time of parent's imminent death** | | | |
| Have experienced being physically assaulted or sexually violated | | | |
| Yes | 28 (37) | 47 (63) | 75 |
| No | 105 (20) | 432 (80) | 537 |
| Have experienced being bullied | | | |
| Yes | 53 (31) | 116 (69) | 169 |
| No | 80 (18) | 361 (82) | 441 |
| Awareness time at which the teenager realized the parent would die from the disease | | | |
| At the time of the death | 30 (24) | 93 (76) | 123 |
| Hours to days before the death | 40 (21) | 154 (79) | 194 |
| Weeks to months before the death | 54 (22) | 188 (78) | 242 |
| 6 months or longer before the death | 8 (16) | 43 (84) | 51 |

[a] Poor = no/little

[b] Good = moderate/ very good.

**Table 2. Characteristics of the study population.**

| Characteristic | n (%) |
|---|---|
| **Participants** | |
| Confirmed eligible [a] | 851 |
| Not reachable | 55 (6) |
| Declined to participate | 66 (8) |
| Agreed but did not return questionnaire | 108 (13) |
| Provided information | 622 (73) |
| **Gender of the participants** | |
| Male | 309 (50) |
| Female | 312 (50) |
| Not stated [c] | 1 |
| **Year of birth** | |
| 1988–1990 | 210 (34) |
| 1986–1987 | 286 (46) |
| 1984–1985 | 123 (20) |
| Not stated [c] | 3 |
| **Birth order** | |
| Oldest child | 144 (23) |
| Middle child | 148 (24) |
| Youngest child | 302 (49) |
| No siblings | 27 (4) |
| Not stated [c] | 1 |
| **Current employment status [b]** | |
| Studying at high school level | 24/614 (4) |
| Adult education at high school level | 31/613 (5) |
| Studying at university level | 187/613 (30) |
| Employed or self-employed | 355/616 (58) |
| Unemployed | 91/616 (15) |
| On parental leave | 9/613 (2) |
| On sick leave | 7/613 (1) |
| **Gender of the deceased parent** | |
| Male | 337(54) |
| Female | 284 (46) |
| Not stated [c] | 1 |
| **Father's year of birth** | |
| 1960–1969 | 60 (10) |
| 1955–1959 | 162 (27) |
| 1950–1954 | 176 (29) |
| 1936–1949 | 203 (34) |
| Not stated [c] | 21 |
| **Mother's year of birth** | |
| 1960–1969 | 125 (21) |
| 1955–1959 | 205 (35) |
| 1950–1954 | 184 (31) |
| 1936–1949 | 78 (13) |
| Not stated [c] | 30 |
| **Father's level of education** | |
| Middle school ($\leq$ 9th grade) | 131 (22) |

(*Continued*)

**Table 2.** (Continued)

| Characteristic | n (%) |
|---|---|
| High school (≥10th grade) | 244 (41) |
| College/university | 215 (37) |
| Not stated c | 32 |
| **Mother's level of education** | |
| Middle school (≤ 9th grade) | 94 (15.9) |
| High school (≥10th grade) | 245 (41.5) |
| College/university | 252 (42.6) |
| Not stated c | 31 |

a Confirmed eligible = all those identified in registers who met the inclusion criteria.

b More than one response alternative could be selected for this question. Number of responses per answer is provided.

c The group "not stated" is not included in calculations of prevalence.

maternally bereaved, after final adjustments for possible confounding factors. In contrast, a higher risk of low well-being and problematic sleeping at the time of the survey remained statistically significant only among the paternally bereaved participants.

## Discussions

The main results of this study show that poor family cohesion, as reported by a parentally bereaved youth, the first year after losing a parent to cancer as a teenager was strongly associated with psychological health-related problems six to nine years after the loss. The health-related problems included symptoms of moderate to severe depression, low well-being, symptoms of anxiety, problematic sleeping and emotional numbness. Our results are in line with

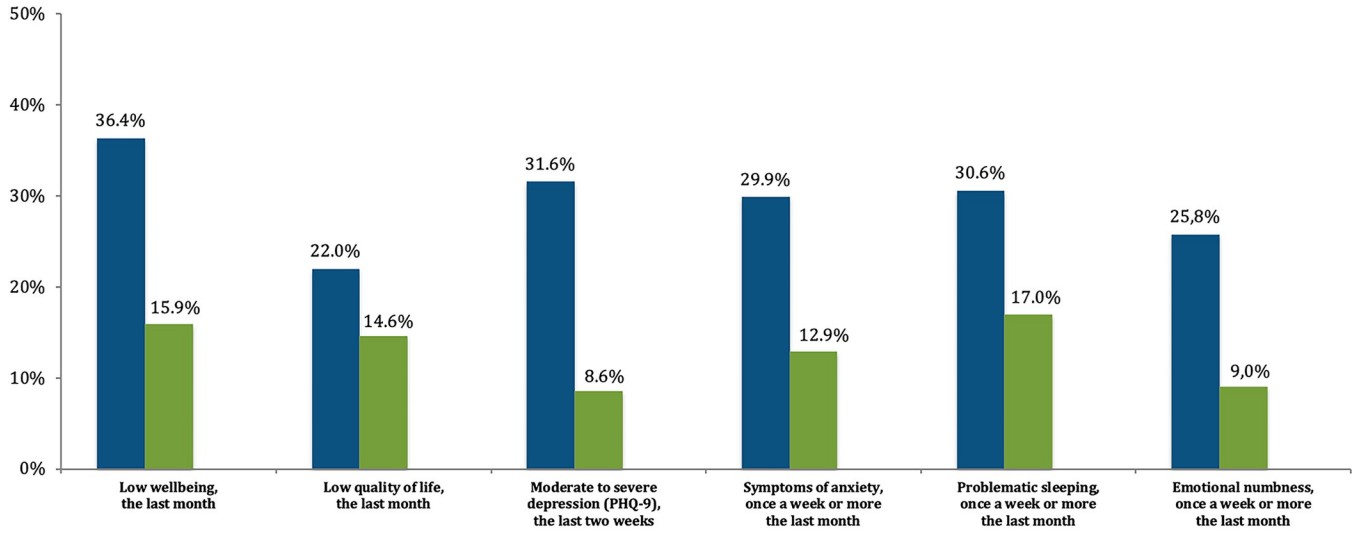

**Fig 1. Prevalence of long-term negative health-related outcomes among cancer-bereaved young adults (six to nine years post loss) in relation to self-reported family cohesion the first year after the death of a parent as a teenager.** Note, for graphical reasons, only the frequencies between 0% and 50% are included.

**Table 3. Self-reported family cohesion the first year after the loss of a parent to cancer and long-term health-related outcomes.**

| Health-Related outcome at time of survey (6–9 years after the loss): | Family cohesion | | | Unadjusted | Adjusted d | | |
|---|---|---|---|---|---|---|---|
| | Poor a | Good b | Missing c | RR | RRAdjusted 1 | RRAdjusted 2 | RRAdjusted 3 |
| | n/total (%) | n/total (%) | n (%) | (95% CI) | (95% CI) | (95% CI) | (95% CI) |
| **Low Well-being the last month** | | | | | | | |
| Whole group | 48/132 (36.4%) | | 8 (1.3%) | **2.28** (1.68–3.10) | **2.04** (1.47–2.83) | **2.06** (1.46–2.92) | **1.89** (1.32–2.71) |
| | | 76/477 (15.9%) | | – | – | – | – |
| Paternally bereaved | 27/61 (44.3%) | | 3 (0.5%) | **2.67** (1.81–3.93) | **2.49** (1.65–3.74) | **2.41** (1.58–3.69) | **2.16** (1.36–3.41) |
| | | 45/271 (16.6%) | | – | – | – | – |
| Maternally bereaved | 21/71 (29.6%) | | 5 (0.8%) | **1.96** (1.21–3.17) | 1.57 (0.90–2.70) | 1.55 (0.87–2.74) | 1.50 (0.85–2.64) |
| | | 31/205 (15.1%) | | – | – | – | – |
| **Low Quality of Life the last month** | | | | | | | |
| Whole group | 29/132 (22.0%) | | 6 (1.0%) | **1.50** (1.02–2.22) | 1.33 (0.87–2.02) | 1.35 (0.87–2.10) | 1.13 (0.73–1.76) |
| | | 70/479(14.6%) | | – | – | – | – |
| Paternally bereaved | 13/61 (21.3%) | | 3 (0.4%) | 1.48 (0.84–2.60) | 1.32 (0.73–2.38) | 1.16 (0.63–2.15) | 0.97 (0.52–1.78) |
| | | 39/271 (14.4%) | | – | – | – | – |
| Maternally bereaved | 16/71 (22.5%) | | 3 (0.4%) | 1.51 (0.88–2.58) | 1.20 (0.66–2.18) | 1.26 (0.68–2.34) | 1.23 (0.67–2.24) |
| | | 31/207 (14.4%) | | – | – | – | – |
| **Moderate/severe depression the last two weeks (PHQ-9)** | | | | | | | |
| Whole group | 42/133 (31.6%) | | 7 (1.1%) | **3.67** (2.50–5.40) | **3.07** (2.02–4.66) | **3.22** (2.07–5.00) | **2.63** (1.67–4.15) |
| | | 41/477 (8.6%) | | – | – | – | – |
| Paternally bereaved | 19/62 (30.6%) | | 6 (1.0%) | **3.56** (2.07–6.11) | **3.41** (1.89–6.11) | **3.79** (1.99–7.24) | **2.63** (1.34–5.15) |
| | | 23/267 (8.6%) | | – | – | – | – |
| Maternally bereaved | 23/71 (32.4%) | | 1 (0.2%) | **3.76** (2.16–6.55) | **2.72** (1.54–4.80) | **3.16** (1.65–6.06) | **2.92** (1.53–5.57) |
| | | 18/209 (8.6%) | | – | – | – | – |
| Health-Related outcome at time of survey (6–9 years after the loss): | Family cohesion | | | Unadjusted | Adjusted d | | |
| | Poor a | Good b | Missing c | RR | RRAdjusted 1 | RRAdjusted 2 | RRAdjusted 3 |
| | n/total (%) | n/total (%) | n (%) | (95% CI) | (95% CI) | (95% CI) | (95% CI) |
| **Symptoms of anxiety once a week or more the last month** | | | | | | | |
| Whole group | 40/134 (29.9%) | | 1 (0.2%) | **2.32** (1.64–3.29) | **2.03** (1.42–2.90) | **2.00** (1.35–2.96) | **1.69** (1.14–2.51) |
| | | 62/482 (12.9%) | | – | – | – | – |
| Paternally bereaved | 20/63 (31.7%) | | 0 (0.0%) | **2.62** (1.61–4.24) | **2.10** (1.27–3.46) | **1.87** (1.07–3.27) | 1.35 (0.73–2.50) |
| | | 33/272 (12.1%) | | – | – | – | – |
| Maternally bereaved | 20/71 (28.2%) | | 1 (0.2%) | **2.03** (1.23–3.35) | **1.80** (1.08–3.00) | **2.14** (1.20–3.82) | **2.10** (1.18–3.75) |
| | | 29/209 (13.9%) | | – | – | – | – |
| **Problematic sleeping once a week or more the last month** | | | | | | | |
| Whole group | 41/134 (30.6%) | | 1 (0.2%) | **1.80** (1.30–2.48) | **1.72** (1.23–2.41) | **1.85** (1.31–2.61) | **1.65** (1.14–2.38) |
| | | 82/482 (17.0%) | | – | – | – | – |
| Paternally bereaved | 24/63 (38.1%) | | 0 (0.0%) | **2.21** (1.47–3.32) | **2.07** (1.33–3.23) | **2.25** (1.41–3.59) | **2.17** (1.28–3.68) |
| | | 47/272 (17.3%) | | – | – | – | – |
| Maternally bereaved | 17/71 (23.9%) | | 1 (0.2%) | 1.43 (0.86–2.39) | 1.24 (0.80–2.24) | 1.46 (0.84–2.53) | 1.14 (0.65–1.98) |
| | | 35/209 (16.7%) | | – | – | – | – |
| **Emotional numbness once a week or more the last month** | | | | | | | |
| Whole group | 34/132 (25.8%) | | 7 (1.1%) | **2.86** (1.91–4.30) | **2.68** (1.73–4.15) | **2.38** (1.45–3.92) | **1.98** (1.20–3.27) |
| | | 43/478 (9.0%) | | – | – | – | – |
| Paternally bereaved | 16/61 (26.2%) | | 3 (0.5%) | **3.23** (1.81–5.78) | **3.00** (1.60–5.60) | **2.31** (1.09–4.90) | 1.96 (0.90–4.26) |
| | | 22/271 (8.1%) | | – | – | – | – |

*(Continued)*

**Table 3.** (Continued)

| Maternally bereaved | 18/71 (25.4%) | | 4 (0.6%) | **2.49** (1.41–4.39) | **2.49** (1.31–4.71) | **2.26** (1.12–4.57) | **2.02** (1.01–4.03) |
|---|---|---|---|---|---|---|---|
| | | 21/206 (10.2%) | | – | – | – | – |

**Abbreviations:** RR: Relative risk ratio; CI: Confidence interval.

[a] Poor = no/little

[b] Good = moderate/ very good—1.00 [reference]

[c] The number and % of missing values for each health-related outcome variable out of the 617 participant that answered the question regarding family cohesion (5 participants did not give an answer to that question or 0.8%). One participant did not state the gender of the deceased parent.

[d]: **Variables added to the modified Poisson regression model at each step to calculate the adjusted RRs**

$RR_{Adjusted\ 1}$: Gender, Year of birth, Age at loss. $RR_{Adjusted\ 2}$: Variables from $RR_{Adjusted\ 1}$ + Number of siblings, Educational level of surviving parent, Depression in at least one parent, Alcohol and/or drug misuse in at least one parent. $RR_{Adjusted\ 3}$: Variables from $RR_{Adjusted\ 1\ \&\ 2}$ + Have experienced being bullied, Have experienced being physically assaulted or sexually violated, Awareness time that participant knew his/her parent would die from the disease.

the growing literature concluding that family environment seems to be a major mediating factor in the health and well-being of bereaved-individuals [7,9,21,40–43]. This study adds to this body of knowledge with results that are based on a long-term follow up.

Regarding anxiety and depression, the results are in harmony with other studies showing that good family function is a protecting factor for psychological well-being in bereaved children [1,7,9,21,40,42]. Well-being has also been shown to be associated with high levels of family cohesion in adolescents in the general population [44].

When it comes to sleeping problems in bereaved youth, the existing literature is very limited. A qualitative study on teenagers in the USA showed that sleep disturbance was a common theme the first year after losing a parent [45]. Studies on bereaved adults have shown shorter duration and lower quality of sleep compared to non-bereaved adults [8,43]. The results of the present study show that the cancer-bereaved participants with poor family cohesion the first year after the loss had almost two times the risk of experiencing sleeping problems once a week or more the in last month compared to those with moderate or good family cohesion after the loss. This result is in line with studies, not related to bereavement, that have shown a strong association between family functioning (including cohesion) and sleeping problems in adolescents [46,47]. Our study does not provide any explanation about the causal link between family cohesion and long-term sleeping problems and more studies are needed to understand the mechanism that lies behind this association.

To the extent of our knowledge, no other study has looked at the association between family cohesion after loss and long-term emotional numbness in this population. However, emotional numbness is a common reaction following loss and can be used as one of the indicators for prolonged grief [48]. Previous studies have found that family function right after the death of a family member can predict the family members' capacity to cope with the loss after the death [49,50]. Although not to be answered with our results, this may give us a reason to speculate if the reported emotional numbness in our study might possibly be an indicator for long term prolonged grief, but that would require further investigation.

Mothers have been shown more likely to adapt better to life after the death of their spouse [51] and to better respond to their children's loss-related needs [52], compared to widowed fathers. Although previous reports based on the same study population showed higher prevalence of reported poor family cohesion among the maternally bereaved participants than the paternally bereaved [53], our results did not give a clear indication of what role the gender of the diseased parent plays on bereaved youths' long-term health and well-being. Although not within the scope of this article, a sensitivity analysis was also undertaken with the data stratified

by both gender of the deceased parent and gender of the participant. We did not proceed further with this analysis owing to loss of power, but based on the prevalence numbers alone we saw that female participants reported higher total prevalence of long-term health-related problems, both among the paternally and the maternally bereaved participants (S2 Table). Although we cannot draw any conclusions, these observations might raise our interest to further study the complex phenomenon of family cohesion, loss, and long-term health among parentally bereaved youth taking gender roles into consideration.

## Methodological considerations

Large sample size and high participation rate (73.1%) are some of the main strengths of this study. Additional strength lies in the self-reported data collected directly from the target group, through a thoroughly prepared study-specific questionnaire, based on qualitative interviews with bereaved teenagers and young adults. Through pilot testing the questions were adjusted and carefully formulated before being face-validated in think-aloud interviews. Moreover, the broad variety of health-related outcome variables strengthened the study by giving a better picture of the overall psychological health and well-being of the participants.

When it comes to the limitations that should be considered when interpreting the results of the study, we can first mention that the study measures complex phenomena and it is possible that the analysis didn't cover all potential confounding factors. It is also uncertain if the findings of this study are generalizable to other groups of bereaved adolescents, for example, those that have lost a parent to other diseases or accidents. Also, not including participants from other family constellations, such as children of single parents or children living with same gender parents and children of first-generation immigrants, further limits the generalizability of the study, that should be highly relevant to look into in future studies. However, these limits were made for practical reasons such as to avoid language barriers or the need to assess an even larger set of other possible confounding factors from i.e., trauma experiences of war or refugee. Also, for ethical reasons, to prevent the possible harm that might arise if some of the participants not living with both parents at the time of death, were unaware of their biological parents' death, before receiving an invitation to the study, as demanded by the register holders.

The cross-sectional design of the study requires the possibility of recall-induced bias to be taken into consideration. However, the option of performing the study with a prospective cohort design was ruled out for ethical and practical reasons. Furthermore, a study investigating the accuracy of reports on family environment showed that collecting data with a retrospective design can be valuable for capturing the emotional dimension of family life, including family cohesion [54].

For most of the questions used in this study we followed the well-established, so-called "one phenomenon–one direct question" approach [37], meaning that the questions are designed to directly address a real-life phenomenon. This approach allowed us to collect a comprehensive set of data from the study participants, based on the subjective experiences of the participants themselves. Self-perceived health and well-being is individual, bound to one's own perspective and feelings [55]. This applies also to the perception of family cohesion, which is similarly influenced by circumstances and culture [44]. A study on adolescents newly diagnosed with cancer and their families showed that adolescents rated family cohesion, communication and adaptability poorer than their parents, highlighting how important it is to involve adolescents when assessing family cohesion [56]. Since much of existing literature has looked at family cohesion from the parents' perspective, this study adds important information to better understand the link between family cohesion and bereavement based on the perspective of the youths themselves.

Not using standardized instruments to measure the level of family cohesion could be viewed as a limitation, since we only measured the participants own subjective perception of the level of family cohesion. Therefore, we cannot know what exactly the concept of family cohesion inhibits for them. However, using a global single-item question can also be considered a strength and be preferred over a multiple-item scale in some cases, especially when the intention is to measure complex phenomena [57]. Using a single item question to measure the complex concept of family cohesion was therefore considered appropriate to serve the intention of this study, which was to measure the participants own subjective perception of the level of family cohesion. Moreover, all the instruments available to measure family cohesion had a large number of items [58] and none of them had been validated for our target group. It should also be noted that the concept of family cohesion was well understood by all of the participants involved in the face-validity interviews. Further studies could be made to gain a deeper knowledge on what the concept of family cohesion means to bereaved teenagers and young adults, the underlying mechanisms of how and why the family cohesion chances after the death of a parent and what factors can support good family cohesion in families facing bereavement.

## Clinical implications

There is a strong need to focus on health promotion within palliative care [59–61], including the prevention of ill health as a result of bereavement. The results of this study show that poor family cohesion the first year after the death of a parent was associated with negative psychological health and well-being long-term. Although more research is needed, the evidence we have today suggests that supportive interventions can be beneficial to families with minor children facing bereavement [30,62]. Interventions shown to be beneficial for children and young people include education about wide range of normative grief responses, support with the recognition and regulation of difficult emotions, and to find ways to reduce distress and strengthen healthy coping strategies [63]. Also, to help the youth create continuing bonds with the deceased parent, engage in meaning making, and see possibilities in a future after the loss of a parent [63–65]. Facilitating parental grief and supporting the bereaved parent to be better equipped to meet the needs of the children and strengthen positive family interactions have also been shown to be important for healthy coping after the loss of a parent [1,63,64].

After an assessment of clinical routines and structures in place when a parent dies, undertaken a few years ago, the Swedish health authorities reported that children facing the death of a parent are often forgotten and that there is a lack of clinical routines to support them [66]. At the same time, studies highlight the importance of children and teenagers being able to prepare for the death of a parent [67–69]. Teenagers also want to be told about the parent's disease prognosis [70] and when the death is imminent [71]. After a systematic review of interventions for bereaved children and adolescents, Kühne et al. (2012) concluded that successful interventions for the bereaved family need to start in palliative care and continue after the death [62]. This recommendation is in line with the conclusion of two systematic reviews stating that even brief interventions may prevent problems in psychological health for children and adolescents, providing they take place early and are aimed at children at higher risk [30,72]. In line with those findings, other studies have highlighted the importance of routinely identifying and supporting high-risk families during the period of illness and after the death [41,73–75] to minimize long-term ill health. The results of the current study can strengthen those findings. If supporting families during the period of parental illness and immediately after the loss will contribute to better family function and cohesion that might improve long-term health and well-being among bereaved-adolescent, then this fact should be highlighted for those working in palliative care and with bereaved families. New efforts could be put in place to create

routines in clinical practice that support health-care personnel identifying when there are minor children in a family facing the loss of a parent and recognizing their need for information and support. Future studies focusing on whether a single item question about family cohesion could be valid to use in clinical practice and during bereavement support, to assist with identifying families that might be in need of further support, could be of interest.

## Conclusion

Self-reported poor family cohesion the first year after the loss of a parent to cancer was strongly associated with a variety of long-term negative outcomes related to psychological health and well-being among bereaved youth. To identify families at risk of poor cohesion and to provide them with the support needed to strengthen family cohesion might be a health prevention worth the effort, possibly preventing long-term suffering in teenage offspring facing the death of a parent.

## Supporting information

**S1 Table. Questions used for evaluating health-related outcomes, response options and categorization.**
(DOCX)

**S2 Table. Prevalence of self-reported family cohesion the first year after the loss of a parent to cancer and long-term health-related outcomes, stratified by gender of the deceased parent and gender of the participant.**
(DOCX)

## Acknowledgments

We would like to express our sincere appreciation to all of the study participants for sharing their experience with us. We would also like to thank Tommy Nyberg and Anton Nilsson for their valuable statistical advice.

## Author Contributions

**Conceptualization:** Dröfn Birgisdóttir, Tove Bylund Grenklo, Ulrika Kreicbergs, Carl Johan Fürst.

**Data curation:** Dröfn Birgisdóttir, Tove Bylund Grenklo.

**Formal analysis:** Dröfn Birgisdóttir.

**Funding acquisition:** Ulrika Kreicbergs, Gunnar Steineck, Carl Johan Fürst.

**Investigation:** Dröfn Birgisdóttir, Tove Bylund Grenklo, Ulrika Kreicbergs.

**Methodology:** Dröfn Birgisdóttir, Tove Bylund Grenklo, Ulrika Kreicbergs, Gunnar Steineck, Carl Johan Fürst, Jimmie Kristensson.

**Resources:** Carl Johan Fürst.

**Supervision:** Tove Bylund Grenklo, Ulrika Kreicbergs, Carl Johan Fürst, Jimmie Kristensson.

**Validation:** Tove Bylund Grenklo, Ulrika Kreicbergs.

**Visualization:** Dröfn Birgisdóttir.

**Writing – original draft:** Dröfn Birgisdóttir.

**Writing – review & editing:** Dröfn Birgisdóttir, Tove Bylund Grenklo, Ulrika Kreicbergs, Gunnar Steineck, Carl Johan Fürst, Jimmie Kristensson.

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
