## [Decision Letter · Decision Letter 0]

3 Nov 2022

PONE-D-22-02706Family cohesion predicts long-term health and well-being after losing a parent to cancer as a teenager: A nationwide population-based study.PLOS ONE

Dear Dr. Birgisdóttir,

Thank you for submitting your manuscript to PLOS ONE. After careful consideration, we feel that it has merit but does not fully meet PLOS ONE’s publication criteria as it currently stands. Therefore, we invite you to submit a revised version of the manuscript that addresses the points raised during the review process. Please address comments made by the reviewers specially regarding concepts of " family composition " and "family cohesion".

We look forward to receiving your revised manuscript.

Kind regards,

Amir Radfar, MD,MPH,MSc,DHSc

Academic Editor

PLOS ONE

Journal Requirements:

We would also like to thank the Swedish Cancer Foundation [2008–758], The Gålö Foundation, The Kamprad Family Foundation for Entrepreneurship, Research and Charity, and The Mats Paulssons Stiftelse for supporting the research project with grants. 

However, funding information should not appear in the Acknowledgments section or other areas of your manuscript. We will only publish funding information present in the Funding Statement section of the online submission form. 

The Swedish Cancer Foundation (2008–758), https://www.cancerfonden.se (GS);

The Kamprad Family Foundation for Entrepreneurship, https://familjenkampradsstiftelse.se/in-english/ (CJ); The Mats Paulsson Foundation, https://www.matspaulssonstiftelserna.com (CJ), and the Gålö Foundation, https://www.galostiftelsen.se (UK) supported the research project. The funders had no role in study design, data collection and analysis, decision to publish, or preparation of the manuscript.

Reviewers' comments:

Reviewer's Responses to Questions

**Comments to the Author**

1. Is the manuscript technically sound, and do the data support the conclusions?

Reviewer #1: Yes

Reviewer #2: Yes

2. Has the statistical analysis been performed appropriately and rigorously? 

Reviewer #1: Yes

Reviewer #2: Yes

3. Have the authors made all data underlying the findings in their manuscript fully available?

Reviewer #1: Yes

Reviewer #2: Yes

4. Is the manuscript presented in an intelligible fashion and written in standard English?

Reviewer #1: Yes

Reviewer #2: Yes

5. Review Comments to the Author

Reviewer #1: The manuscript “Family cohesion predicts long-term health and well-being after losing a parent to cancer as a teenager: A nationwide population-based study” is very well written, the results are very well presented, the statistical analysis is well done, and it will potentially promote preventive interventions in bereaved teenagers.

However, the concept of family has been changing during the last decades. IMHO the authors started from a traditional family concept without considering different families, socio-economical and educational levels, cultural and religious backgrounds etc.

In addition, the composition of families was not described at all. The concept of living together when the parent dies is an administrative requirement, but what about brothers and sisters, grandparents, etc.? There are no specifications.

Consequently, the definition of family cohesion should be described to understand what we are discussing as a potentially protected factor.

I highly recommend considering these questions face to future generations of family members. I suggest briefly defining the “family” you managed and why. Furthermore, what do you consider “family cohesion” to mean? You also should include some discussion about these concepts and your results.

Reviewer #2: Parentally bereaved children are an underserved population. This study is important as it adds information to help healthcare professionals understand the need for family support after the death of a parent and highlights the long term impact of parental loss during during adolescence.

The only question I have concerns using the cutoff of 16 versus 18 years old for age at the time of parental death. Why not include adolescents who had lost a parent between the ages of 13-18 years old?

6. PLOS authors have the option to publish the peer review history of their article (what does this mean?). If published, this will include your full peer review and any attached files.

Reviewer #1: **Yes: **Vilma Adriana Tripodoro

Reviewer #2: No

---

## [Author Response · Author response to Decision Letter 0]

8 Feb 2023

Response to reviewers

Editors comment #1

Authors response to editors’ comment #1

Dear editor we have now updated our documents in accordance to your style templates. 

Editors comment #2

You indicated that you had ethical approval for your study. In your Methods section, please ensure you have also stated whether you obtained consent from parents or guardians of the minors included in the study or whether the research ethics committee or IRB specifically waived the need for their consent.

Authors response to editors’ comment #2

Dear editor, no minors were included in the study and all of the participants where adults at the time of participation (18 to 26 years of age), therefore no consent was needed from their parents who no longer were considered to be their guardians at the time of the study according to Swedish laws. 

Editors comment #3

Thank you for stating the following in the Acknowledgments Section of your manuscript: 

We would also like to thank the Swedish Cancer Foundation [2008–758], The Gålö Foundation, The Kamprad Family Foundation for Entrepreneurship, Research and Charity, and The Mats Paulssons Stiftelse for supporting the research project with grants. 

However, funding information should not appear in the Acknowledgments section or other areas of your manuscript. We will only publish funding information present in the Funding Statement section of the online submission form. 

The Swedish Cancer Foundation (2008–758), https://www.cancerfonden.se (GS);

The Kamprad Family Foundation for Entrepreneurship, https://familjenkampradsstiftelse.se/in-english/ (CJ); The Mats Paulsson Foundation, https://www.matspaulssonstiftelserna.com (CJ), and the Gålö Foundation, https://www.galostiftelsen.se (UK) supported the research project. The funders had no role in study design, data collection and analysis, decision to publish, or preparation of the manuscript.

Authors response to editors’ comment #3

Dear editor we have now removed all funding information from the section additional information in the manuscript (line 865 in the Revised Manuscript with tracked changes). The information in the funding statement is correct and not in need of an update and therefore no amended statements regarding this are included in the cover letter. 

Editors comment #4

We note that you have indicated that data from this study are available upon request. PLOS only allows data to be available upon request if there are legal or ethical restrictions on sharing data publicly. For more information on unacceptable data access restrictions, please see http://journals.plos.org/plosone/s/data-availability#loc-unacceptable-data-access-restrictions. 

Authors response to editors’ comment #4

Dear editor we have now removed the data availability statement from the Additional Information section of the manuscript (see line 865 in the Revised Manuscript with tracked changes). We have now updated the data availability statement of the manuscript, the new statement is written as shown here below: 

In order to assure data confidentiality and to protect the privacy of the research participants, the datasets generated and/or analyzed during the current study are not publicly available due to legal and ethical restrictions as described by the Swedish law (2003:460) and the Swedish Ethical Review Authority (https://etikprovningsmyndigheten.se) regarding processing of sensitive data. Data can be made available from the Department of Oncology‐Pathology, Division of Clinical Cancer Epidemiology, Karolinska Institute, Stockholm, Sweden (contact via gunnar.steineck@ki.se), for researchers who meet the criteria for access to confidential data.

Editors comment #5

Your ethics statement should only appear in the Methods section of your manuscript. If your ethics statement is written in any section besides the Methods, please delete it from any other section. 

Authors response to editors’ comment #5

We have now removed the ethical statement that was listed under the section Additional information in the manuscript. (See line 865) and moved it to the Methods section (see lines 144–150)

Editors comment #6

Please include captions for your Supporting Information files at the end of your manuscript, and update any in-text citations to match accordingly. Please see our Supporting Information guidelines for more information: http://journals.plos.org/plosone/s/supporting-information. 

Authors response to editors’ comment #6

We have now added a Supporting information heading and caption for our Supporting information files in lines 773–777 and we have updated the in-text citations, see lines 197 and 482 in the Revised Manuscript with tracked changes.

Editors comment #7

Authors response to editors’ comment #7

Thank you we have now reviewed our reference list and made the following changes: 

o Reference nr. 1 has been updated and now cites the publisher's final edited version of the article. 

o Reference nr. 21 appeared two times in the reference list (ref. nr. 21 and 39) in the previous submission of the manuscript. The duplicate has now been removed.

o References nr 24 and 25 are new to the reference list, cited in line 105 and 106 in the revised manuscript.

o Reference 29, available from link has now been added to the reference.

o Reference nr. 33 is new to the reference list, cited in line 135 in the revised manuscript.

o Reference nr. 58 is new to the reference list, cited in line 546 in the revised manuscript.

o Reference 67, available from link has now been added to the reference.

o Available DOI, PMID and PMCID have been added to the references in the reference list, when applicable.

Reviewer #1 - Comments

The manuscript “Family cohesion predicts long-term health and well-being after losing a parent to cancer as a teenager: A nationwide population-based study” is very well written, the results are very well presented, the statistical analysis is well done, and it will potentially promote preventive interventions in bereaved teenagers.

However, the concept of family has been changing during the last decades. IMHO the authors started from a traditional family concept without considering different families, socio-economical and educational levels, cultural and religious backgrounds etc.

In addition, the composition of families was not described at all. The concept of living together when the parent dies is an administrative requirement, but what about brothers and sisters, grandparents, etc.? There are no specifications.

Consequently, the definition of family cohesion should be described to understand what we are discussing as a potentially protected factor.

I highly recommend considering these questions face to future generations of family members. I suggest briefly defining the “family” you managed and why. Furthermore, what do you consider “family cohesion” to mean? You also should include some discussion about these concepts and your results.

Authors response to reviewers’ comments

Dear reviewer we would like to thank you for taking the time to review our manuscript and for your useful comments to our manuscript. 

Regarding family cohesion: 

The definition of family cohesion that was given in the background section of the manuscript has now been updated (see lines 110–111 in the Revised Manuscript with tracked changes). 

The aim of the study was to measure the participants own subjective experience of family cohesion and was measured with a single item question as described in the method section. This means that we cannot know what the phenomenon of family cohesion means to each participant since we have only measured their own perception of the level of family cohesion. We have now clarified this in lines 538–545 in the Revised Manuscript with tracked changes.

We have also suggested further research to probe deeper into the concept of family cohesion in lines 548–552 and under clinical implications in lines 595–598 in the Revised Manuscript with tracked changes.

Regarding the family construction: 

Regarding the reviewers’ quest for better defining the family, we have now added in the Introduction section a text that describes the concept of family based on Bowens family systems theory (see lines 103–106 in the Revised Manuscript with tracked changes). 

Since this was a nationwide population-based study, all those who had lost a parent to cancer during their teenage years in the reference years and who had been living in a two parents’ household were invited to participate, this included people from all geographical areas within the country of Sweden and of different socioeconomic statuses. Additionally, the participants also needed to be born in one of the Nordic countries and speak and understand Swedish. 

Factors such as income, education and occupational status are often used to indicate the socioeconomic status. We did not collect any data on the participants or their parents’ income level, but in table 2 the educational level of the parents and the participants current employment status, are stated, which can give an indication of the socioeconomic status. An information about the number of siblings of the participants is also shown in table 1, but information about other social circumstances, such as support from grandparents or religious background was not within the scope of this paper. 

To address why first-generation immigrants’ and other forms of family constellations than two-parents’ households were excluded from the study, has now been clarified in the methodological discussion, lines 504–513 in the Revised Manuscript with tracked changes.

Reviewer #2: - Comments

Parentally bereaved children are an underserved population. This study is important as it adds information to help healthcare professionals understand the need for family support after the death of a parent and highlights the long-term impact of parental loss during adolescence.

The only question I have concerns using the cutoff of 16 versus 18 years old for age at the time of parental death. Why not include adolescents who had lost a parent between the ages of 13-18 years old?

Authors response to reviewers’ comments

Dear reviewer we would like to thank you for your comments to our manuscript. We have now clarified the rationale behind restricting the age of the participants to 13-16 years of age at the time of death (see lines 132–135 in the Revised Manuscript with tracked changes).

---

## [Decision Letter · Decision Letter 1]

7 Mar 2023

Family cohesion predicts long-term health and well-being after losing a parent to cancer as a teenager: A nationwide population-based study.

PONE-D-22-02706R1

Dear Dr. Birgisdóttir,

We’re pleased to inform you that your manuscript has been judged scientifically suitable for publication and will be formally accepted for publication once it meets all outstanding technical requirements.

Kind regards,

Amir Radfar, MD,MPH,MSc,DHSc

Academic Editor

PLOS ONE

Additional Editor Comments (optional):

Reviewers' comments:

Reviewer's Responses to Questions

**Comments to the Author**

1. If the authors have adequately addressed your comments raised in a previous round of review and you feel that this manuscript is now acceptable for publication, you may indicate that here to bypass the “Comments to the Author” section, enter your conflict of interest statement in the “Confidential to Editor” section, and submit your "Accept" recommendation.

Reviewer #1: All comments have been addressed

Reviewer #2: All comments have been addressed

2. Is the manuscript technically sound, and do the data support the conclusions?

Reviewer #1: Yes

Reviewer #2: Yes

3. Has the statistical analysis been performed appropriately and rigorously? 

Reviewer #1: Yes

Reviewer #2: Yes

4. Have the authors made all data underlying the findings in their manuscript fully available?

Reviewer #1: Yes

Reviewer #2: Yes

5. Is the manuscript presented in an intelligible fashion and written in standard English?

Reviewer #1: Yes

Reviewer #2: Yes

6. Review Comments to the Author

Reviewer #1: The authors have been adressed all querries. The limitations and future researchs were considered.

Congratulations for the article.

Reviewer #2: Children who lose a parent to cancer are an underserved population and this article fills an important void. The authors have addressed the concerns/suggestions.

7. PLOS authors have the option to publish the peer review history of their article (what does this mean?). If published, this will include your full peer review and any attached files.

Reviewer #1: **Yes: **Vilma Adriana Tripodoro

Reviewer #2: No

---

## [Editor Report · Acceptance letter]

20 Mar 2023

PONE-D-22-02706R1 

Family cohesion predicts long-term health and well-being after losing a parent to cancer as a teenager: A nationwide population-based study. 

Dear Dr. Birgisdóttir:

I'm pleased to inform you that your manuscript has been deemed suitable for publication in PLOS ONE. Congratulations! Your manuscript is now with our production department. 

Kind regards, 

on behalf of

Dr. Amir Radfar 

Academic Editor

PLOS ONE